# D2PO: Discriminator-Guided DPO
# with Response Evaluation Models

**Prasann Singhal**[♡], **Nathan Lambert**[♠], **Scott Niekum**[♣], **Tanya Goyal**[◇], **Greg Durrett**[♡]

[♡]The University of Texas at Austin, [♠]Allen Institute for Artificial Intelligence
[♣]University of Massachusetts Amherst, [◇]Princeton University

prasanns@cs.utexas.edu

## Abstract

Varied approaches for aligning language models have been proposed, including supervised fine-tuning, RLHF, and direct optimization methods such as DPO. Although DPO has rapidly gained popularity due to its straightforward training process and competitive results, there is an open question of whether there remain practical advantages of using a discriminator, such as a reward model, to evaluate responses. We propose D2PO, discriminator-guided DPO, an approach for the online setting where preferences are being collected throughout learning. As we collect gold preferences, we use these not only to train our policy, but to train a discriminative response evaluation model to silver-label even more synthetic data for policy training. We explore this approach across a set of diverse tasks, including a realistic chat setting, and we find that our approach leads to higher-quality outputs compared to DPO with the same data budget, and greater efficiency in terms of preference data requirements. Furthermore, we show that our silver labeling is most helpful when training the policy with DPO, outperforming traditional PPO, and benefits from maintaining a separate discriminator from the policy model.

## 1 Introduction

Learning from human preferences is the prevailing method for large language model (LLM) alignment, including approaches like reinforcement learning from human feedback (RLHF) (Ouyang et al., 2022; Bai et al., 2022), Direct Preference Optimization (DPO) (Rafailov et al., 2023), and several recent alternatives (Azar et al., 2023; Ethayarajh et al., 2024; Hong et al., 2024). A key idea in this line of work, introduced by DPO, is that the reward objective can be expressed in terms of the optimal policy and reference policy, allowing us to train a model from preference data without learning a separate reward model or sampling from the policy during learning.

However, the theoretical guarantees of DPO may not apply in practice. Preferences are not necessarily obtained over a set of outputs that are in-distribution for the final aligned model. For example, preferences may be labeled over initial model outputs of a certain length, but the distribution of the policy model may shift during training to produce longer responses (Singhal et al., 2023). In this case, the preference data does not uniquely specify an optimal policy, so different algorithms may lead to different solutions in practice. At the same time, recent approaches have explored collecting preferences from the shifting distribution of the model throughout training (Touvron et al., 2023; Yuan et al., 2024b). It is not clear theoretically nor empirically which approach is best given a limited budget for preference labeling in this online setting.

This paper investigates the role of an explicit discriminative model during the alignment process. Our central hypothesis is that when preference data is limited, a model discriminatively trained to evaluate responses (like a reward model) can learn to assess them more

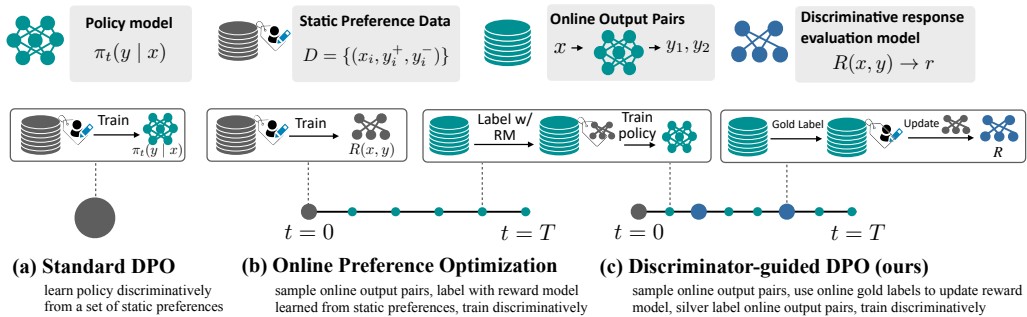

Figure 1: Comparison of standard DPO, online preference optimization methods (with reward model-labeled data), and our proposed D2PO method. The key addition in (c) is the online learning of the reward model on new preferences during policy optimization.

easily than a model can learn to produce them. This discriminative response evaluation model can then be used to silver-label samples from our policy to give additional data for policy training.

We operationalize this approach in a a method called D2PO (Figure 1). D2PO alternates between two phases: first, collecting preference labels to train the discriminative response evaluation model (blue), and second, using that discriminator to label a larger number of outputs from the policy model (teal). We use the discriminator as a *pairwise* preference labeler and train the policy using the DPO objective, which we find works better than a conventional PPO setup (Schulman et al., 2017). Finally, by decoupling the discriminator from the policy, it can be learned over human-labeled preference data only, while the policy model can learn from a larger amount of noisily-labeled on-policy data.

Our results study both a realistic chat benchmark (UltraFeedback (Cui et al., 2023)) as well as several simple text generation tasks where we explicitly define gold reward functions. We show several key findings. First, we show that the online preference collection setting indeed works better than having only static preferences collected from the initial policy model. We then show that, given the same budget of preference updates as baselines, D2PO on several settings achieves higher reward more quickly than an online version of DPO and than basic PPO. Finally, we study how the discriminative reward evaluation model behaves over the course of training. Receiving new labeled data is crucial for it to be able to make accurate judgments about new sampled responses. Combined with controlled D2PO experiments testing different types of discriminators, including using the policy as its own discriminator, we establish that in the setting we discuss, a separate discriminator may still be a useful ingredient in LLM alignment.

Taken together, we make the following contributions. (1) We propose a new approach, discriminator-guided DPO, for the setting where preferences are being collected online. (2) We show, on a diverse set of tasks, that maintaining a discriminative response evaluation model and using it to silver label new sampled responses improves policy training. (3) We release code [1] and our diverse task settings to support the testing and development of future alignment algorithms.

## 2 Background / Setup

Let $\pi(y \mid x)$ be an LM that places a probability distribution over a response $y$ given an input $x$. Let $D = \{(x_i, y_i^+, y_i^-)\}$ be a dataset of human preference judgments, with outputs $y_i^+$ preferred to alternatives $y_i^-$. These preferences are assumed to be derived from a scalar reward model $R^*(x, y)$ reflecting human utility, which we do not have access to. The goal of LM alignment training is to maximize this reward, optionally starting from a supervised fine-tuned policy $\pi_{\text{sft}}(y \mid x)$.

---

[1]Code available at `https://github.com/PrasannS/d2po`

In standard RLHF (PPO; (Schulman et al., 2017)), first, a Bradley-Terry reward model $R(x, y)$ is learned using preference data. To train the policy, **online** rollouts are then iteratively sampled from the policy $\pi_t$ at each training iteration, and the loss optimization depends on the reward scores assigned to these outputs by the learned reward model, allowing the policy to learn from these rollouts under the assumption that the reward model produces accurate rewards. Direct Preference Optimization (DPO; Rafailov et al. (2023)), another popular approach, trains the policy directly, **offline**, on just the initial preference data using a discriminative loss, removing the separate discriminator. However, later work (Liu et al., 2023; Lee et al., 2024), has explored incorporating online sampling with discriminative objectives. We broadly call this work **online preference optimization (OPO)** as a unified template to capture the core design choices of this prior work for comparison to D2PO (defined more in Section 3).

While such methods are online with respect to the policy, they are not online with respect to preferences, which are usually collected from a fixed distribution such as $\pi_{\text{sft}}$. Thus, as distribution $\pi_t$ changes over training, the signal of a reward model or a discriminator may degrade in quality, limiting improvement. Some work (Touvron et al., 2023; Guo et al., 2024) explores collecting preferences at an additional intermediate training step to address this issue. However, the intermediate steps, data, and objectives for this process are often chosen in an ad-hoc manner.

## 3 D2PO**: Using a Discriminator in Policy Training**

**Our setting** This work focuses on the setting where we collect additional gold preference judgments during policy learning. We define the total gold-labeled **preference budget**, typically from humans or from LLM-as-a-judge, as $P$. In most approaches, like DPO and OPO, all preferences $D_0 = \{(x_i, y_i^+, y_i^-)\}$ are collected offline ($P = |D_0|$) from a single initial distribution. In iterative versions (OPO) of DPO, including our proposed method D2PO, gold preference data is collected at $T_P$ different stages during training. The total preference data is therefore $D : \{D_0, D_1 \cdots D_{T_P}\}$, and $P = \sum_{t:0:T_P} |D_t|$.

We hypothesize that **online updates** to discriminators, with a high number $T_P$ of preference collection steps, will help solve the distribution shift issue and improve performance. However, as preference data may be expensive, we want to do so with minimal $P$. The question then becomes how to use a small $P$ to maximally improve our policy with respect to $R^*$. To address this, D2PO combines two ideas: (1) the comparative advantage in data efficiency of online data; (2) the ability of discriminators to generalize quickly on new data and help silver-label data for a policy, reducing the need for costly gold preferences.

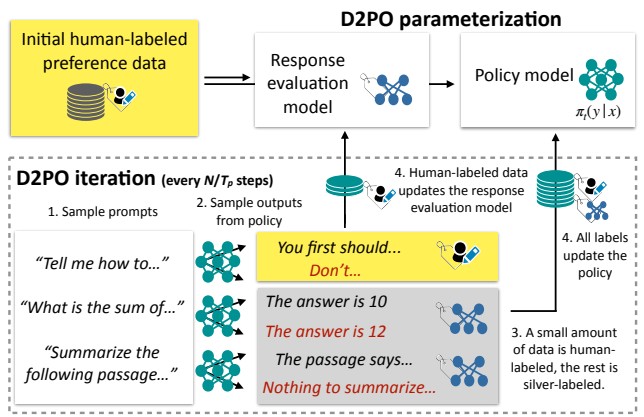

Figure 2: D2PO trains an initial policy model and response evaluation model from gold preferences. It then samples prompts, samples outputs of those prompts, and uses a mix of human labeling and silver labeling to produce policy training data. Only human-labeled data is used to update the response evaluation model.

**Algorithm** We detail our approach, D2PO, in Algorithm 1 and Figure 2. The algorithm, from an initialized policy and discriminator, runs for $N$ iterations. In each iteration, the first step (L3-8) is to sample a batch of paired outputs from the policy $\pi_t$, then get preference labels with our discriminator, and do DPO updates on these new silver preferences. Note this part of the algorithm directly corresponds to several baselines we will compare against: if we omit the rest of the algorithm, this on its own is OPO with a static RM (see below). If we

---

**Algorithm 1** D2PO

---

**Input:** Policy model $\pi$, Discriminator $R$, prompt set $X$, gold preference source $R^*$, hyperparameters: policy rollout budget $N$, gold preference budget $P$

1: $D_{\text{candidates}} \leftarrow \{\}$
2: **for** $t \in [0, N]$ **do**
3:      // Update policy using online rollouts and preferences derived from discriminator $R$.
4:      sample $x \in X$    // sample prompt
5:      $y_1, y_2 \leftarrow \pi(x), \pi(x)$    // get 2 rollouts from policy using prompt
6:      $D_{\text{candidates}} \leftarrow D_{\text{candidates}} \cup (x, y_1, y_2)$
7:      $y^+, y^- \leftarrow \arg\max_{y \in (y_1, y_2)} R(x, y), \arg\min_{y \in (y_1, y_2)} R(x, y)$    // get silver labels from $R$
8:      $\pi \leftarrow \pi + \nabla L_{DPO}(\pi, y^+, y^-)$    // DPO update with new preferences
9:      **if** $t \pmod{N/T_p} == 0$ **then**
10:          // subsample from $D_{\text{candidates}}$, gold preferences and update discriminator R.
11:          $D_{\text{selected}} \leftarrow \arg\text{top } k_{(x,y_1,y_2) \in D_{\text{candidates}}} -|R(x, y_1) - R(x, y_2)|$, where $|D_{\text{selected}}| = P/T_P$
12:          **for** $d \in D_{\text{selected}}$ **do**
13:            $d_g \leftarrow R^*(d)$    // label preference with annotator with gold annotator $R^*$
14:            $R \leftarrow R + \nabla L_{BT}(R, d_g)$    // update reward model
15:            $\pi \leftarrow \pi + \nabla L_{DPO}(\pi, d_g)$    // policy is also updated with new labels
16:          **end for**
17:          $D_{\text{candidates}} \leftarrow \{\}$    // Reset output samples
18:      **end if**
19: **end for**

---

then replace $R$ with $R^*$ here, getting gold labels instead, this becomes the OPO with gold baseline. Lastly, if instead of using $\pi_t$ we fix the rollout distribution to be $\pi_{\text{sft}}$, this gives us DPO.

Every $N/T_p$ steps, we collect gold annotation and update the discriminator (L11-17). In practice we can update the policy with these labels, though this is a sparse set of steps since the number of preferences $P$ is much less than the number of policy training iterations $N$. This is the step depicted in Figure 2. The subsampling step of $D_{\text{candidates}}$ can use a strategy such as confidence sampling (Lewis & Gale, 1994), shown here, where we choose preference pairs with the lowest gap in discriminator reward, or random sampling, where we randomly select $P$ pairs. We generally find these to perform similarly. We report with confidence sampling in most settings, except our Contrastive Distillation setting where we find random sampling to work better.

Importantly, we then update our reward model with these preferences, allowing us to collect automatic preference labels on a larger set of rollouts for "free" (in terms of annotation cost), taking advantage of the discriminator's ability to generalize to other examples from a similar distribution. Note that our algorithm is flexible in the form of the discriminator, which is treated as a black box. We use a Bradley-Terry reward model by default, but in Section 6.2, we investigate if the DPO policy model itself or a separate copy of it can be used as the discriminator instead.

The primary advantage of this approach lies in its efficiency: using purely gold rollouts can be effective, but also slow. By updating the reward model sparsely, these updates can generalize to new examples on the training distribution at each step, allowing more efficient collection and usage of preferences with equivalent performance.

## 3.1 Baselines

To test our hypotheses, we compare our approach to several baselines, including a detailed breakdown of the data conditions for different approaches in Table 1 (corresponding to Figure 3). Note that we hold total gold preferences used, including those used for initialization (Appendix A), constant across all approaches within each setting, though total number of preferences (gold + silver labeled using the discriminator) may vary by setting.

| | GOLD PREFS (OFFLINE) | GOLD PREFS (ONLINE) | SILVER POLICY PREFS (ONLINE) | DISCRIM MODEL | POLICY LOSS | DISCRIM UPDATES (ONLINE) | |
|---|---|---|---|---|---|---|---|
| | | | | | | GOLD | SILVER |
| PPO (static RM) | 2.6k/5.6k/3.6k/3.6k | 0 | 64k | RM | PPO | NO | NO |
| DPO | 2.6k/5.6k/3.6k/3.6k | 0 | 0 | – | DPO | NO | NO |
| OPO (static RM) | 2.6k/5.6k/3.6k/3.6k | 0 | 64k | RM | DPO | NO | NO |
| OPO (gold) | 1.6k | 1k/4k/2k/2k | * | – | DPO | NO | NO |
| D2PO (ours) | 1.6k | 1k/4k/2k/2k | 16k/64k/64k/32k | RM | DPO | YES | NO |
| w/ DPO discrim | 1.6k | 1k/4k/2k/2k | 16k/64k/64k/32k | DPO | DPO | YES | NO |
| w/ Self discrim | 1.6k | 1k/4k/2k/2k | 16k/64k/64k/32k | policy (DPO) | DPO | YES | YES |
| w/ PPO update | 1.6k | 1k/4k/2k/2k | 16k/64k/64k/32k | RM | PPO | YES | NO |

Table 1: Summary of our data and model conditions. Total number of preferences constant across approaches (except in experiments where we explicitly state otherwise), while varying silver data, policy loss function, type of discriminator, and whether discriminator is updated or not. Slashes indicate preference sizes across our Unique Nouns, Word Collector, Contrastive Distillation, and Math settings, respectively.

We first compare against approaches using a static set of preferences labeled over outputs from the initial policy. We compare **DPO**, **PPO (static RM)** and **OPO (static RM)**, the latter two of which train a reward model over the preference data and use it during training, either as a reward in PPO or a labeling function in OPO (see Figure 1). Note that although we introduce the term OPO in this work, the implementation is similar to Liu et al. (2023). The primary difference between OPO and PPO is OPO's use of the DPO loss function over the online examples. We further compare these approaches in a higher data (50K) gold preference setting to examine behavior with many more preferences (Table 2).

We also compare against **OPO (gold)**, where only online gold preference labels are used to update the policy, without any silver labeling. To make the number of policy updates more comparable with D2PO, we train each batch with the DPO objective for 4 epochs; we tuned this hyperparameter to optimize performance (Appendix A).

Finally, we compare against variants of our approach that use both online gold and silver labels, including a version where we use an independently-optimized **DPO discriminator** for our response evaluation model and a version where we use the policy itself (**self discriminator**). We view this latter case as updating both the policy as well as the response evaluation model with silver labels. Finally, we compare against a version where we use PPO updates given our reward model; this is simply PPO with a reward model being updated online.

Our methods have two notions of training progress, following Figure 1: how many online gold preferences we have used from our budget $T_P$ and how many silver updates have been applied to our policy $T_N$. Since human preferences are more costly, we're mainly interested in $T_P$, where we'll hold x-axes constant across approaches with respect to preferences used.

# 4 Experimental Setup

## 4.1 Tasks

To evaluate our policy optimization methods, we evaluate on a diverse set of tasks with distinct $R^*$ that we can derive relative preferences from. Example questions and outputs are in Appendix D). We use four synthetic tasks, designed to exhibit different properties, where we know the ground truth reward function $R^*$, plus one realistic task with two $R^*$ settings.

**Word Collector:** We compute the top 30 content words in the UltraFeedback (Cui et al., 2023) dataset. Given an UltraChat prompt (Ding et al., 2023), the goal is to generate a 50 word output with as many of these words as possible, where the presence of each word gives +1 reward, for a maximum of 30. This simulates having different sparse "positive" features that naturally may not occur together, and learning to incorporate multiple in single outputs. We subsample prompts initial preference dataset from UltraFeedback. We find that this task allows for realistic outputs that achieve high reward, while at the same time being challenging to optimize and showing variation among the training methods we compare.

**Unique Nouns:** Instead of maximizing word coverage, here we maximize the number of unique nouns, detected with spaCy (Honnibal et al., 2020) in a 50 token output. This is an example of a much denser reward function. We get our initial preference dataset by subsampling from UltraFeedback. This reward function is the most easily optimized of those we consider and helps measure how quickly policy learning can be "steered" by the reward.

**Contrastive Distillation:** Given a larger $\theta_L$ (OPT-1.3b, Zhang et al. (2022a)) and smaller model $\theta_S$ (OPT-125m), the $R^*$ is the difference of log probabilities $\log(p(y|x; \theta_L)) - \log(p(y|x; \theta_S))$; we receive more reward for sequences likely under the large model and unlikely under the small model. This task is represents a likelihood-based ground truth reward, similar to how DPO's implicit reward is constructed. We get our initial preference dataset and prompts with OPT 125m samples on truncated (5-15 tokens) Wikipedia data (Foundation).

**Math Expressions:** The input is a mathematical expression randomly generated as a tree of up to two layers deep. From here, there is a single gold sequence, which is a chain-of-thought style sequence where the deepest, left-most sub-expression is solved one at a time (e.g., "((5 + 1) * 2) = (6 * 2) = 12"). Then, given a prediction such as "((5 + 1) * 2) = (6 * 2) = 13", we can iterate through each step, where $R^*$ is -1 multiplied with the total edit distance between solution and predictions at each step. This task has a single solution. Prompts and preferences are randomly generated.

**UltraFeedback:** Finally, for a realistic setting where a "true" ground truth reward is unknown, we use the popular UltraFeedback (Cui et al., 2023) dataset, where we can use their GPT-4 based (`gpt-4-0613`) labeling scheme which returns a score between 1-5 as our $R^*$. We just use subsampled UltraFeedback for our initial data. We further run an alternate setting using EurusRM (Yuan et al., 2024a), the best reward model on RewardBench (Lambert et al., 2024) at the time the work was conducted, as an alternative gold reward.

## 4.2 Implementation

We build on top of the Huggingface TRL framework (von Werra et al., 2020) with hyperparameters we find to work best based on reward convergence and downstream evaluation: $\lambda = 0.05$, batch size 64; more details in Appendix A. We use Llama-2-7B models as our base for realistic experiments, OPT-1.3b for the math setting, and OPT-125m for our synthetic experiments (Zhang et al., 2022a), and use LoRA (rank= 16) (Hu et al., 2021) to enable training with a smaller GPU footprint, finding it to not affect initial experimental DPO/RM results. We use the Tülu SFT models for UltraFeedback (Ivison et al., 2023), an OPT-1.3b model fine-tuned on 500k example math expressions, and OPT-125m for other experiments. For each task, we evaluate the reward over the timesteps of training. Periodically throughout learning, we use our policy to compute gold reward over a held-out dev set of prompts.

## 5 Results

### 5.1 Comparing loss objectives (DPO vs PPO)

Before evaluating the potential of our full D2PO model, we first compare three different baseline approaches in offline settings, i.e., using only initial preference data or static discriminators. We compare: (1) standard DPO, (2) OPO w/ static RM, which uses the DPO loss for paired policy updates, and (3) PPO w/ static RM, i.e., standard PPO. We use a large preference dataset (50K examples) for these experiments.

|  | NOUN | WC | CDIST | MATH |
|---|---|---|---|---|
| DPO | 12 | 6.9 | 0.08 | -17.1 |
| PPO (static RM) | 24.9 | 6.7 | -0.20 | -13.2 |
| OPO (static RM) | 33.2 | 7.8 | 0.22 | -3.5 |

Table 2: Comparison of DPO, OPO and PPO with $P_0 = 50K$. OPO w/ static RM performs best amongst these baselines.

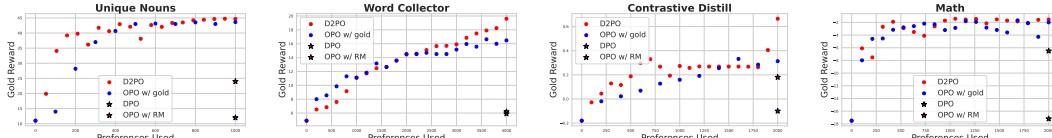

Figure 3: Amount of gold preference data (x-axis; corresponds to progress through training, not counting initial 1.6k offline prefs) vs. resulting gold reward, averaged over 3 runs. We compare D2PO against OPO with gold data only, as well as "basic" DPO and OPO with an RM trained on initial data (note that this is a smaller set than in Table 2). Our method reaches higher reward in Word Collector and Contrastive Distillation, and maxes out faster at Unique Nouns.

**Results:** Table 2 reports the gold final reward of training runs for the three baseline approaches. We find that **OPO w/ static RM outperforms standard DPO** in all settings, even though these use the same loss objective. This highlights the benefits of sampling online rollouts during policy training, as opposed to using a fixed preference set. Our results also show that PPO w/ static RM is generally outperformed by OPO, and even standard DPO for two out of four settings. We generally find the DPO objective to be more stable than PPO, and use the former for all our experiments. Finally, these results give lower reward than we will see in Section 5.2 from using online rewards, showing how effective online reward updates are.

## 5.2 Comparing D2PO against baselines

**Setup:** To compare these for equivalent gold preferences, we report model performance quantified by the gold reward on 200 held-out datapoints (y-axis) against the number of gold preferences used for training (x-axis). That is, we report standard DPO and OPO w/ static RM performance on the same preference budget as the total budget over all time steps of our online D2PO and OPO w/ gold methods. Note, however, that the number of policy updates may differ between these approaches (see Table 1). We report aggregated results across 3 seed runs.

**Results on synthetic tasks:** We compare our proposed approach D2PO and baselines in Figure 3. First, we importantly note that prior methods like DPO and OPO (stars), which are **offline** *with respect to preferences*, **do much worse than the online preference approaches**, even with much more data. Next, comparing the online approaches, we find that on three settings, D2PO **leads to overall improvements in efficiency of data**, either reaching higher final reward or higher intermediate rewards using a lower preference budgets.

For instance, on Word Collector, D2PO reaches a reward score of ~ 35 with a preference budget of $P = 100$ where OPO w/ gold requires $P = 300$ to give the same performance. This suggests that silver-labelling using an iteratively improving RM is an effective strategy to offset the high annotation costs of online gold annotations. Note that this is not as clear on the Math setting, suggesting efficiency gains may depend on the setting. We explore this more in Section 6. Overall, our results suggest that both online preferences and an on-distribution RM can contribute to increased performance and data efficiency.

**Results on realistic task:** Next, we report results on our realistic settings on UltraFeedback in Figure 4. On the GPT-4 annotation-based setting (UltraFeedback), we compare approaches assuming access to a small preference budget of 500 online preferences; gold annotation in this setting using GPT-4 costs $60 per run. We also report results when we use a budget of 3k preferences on the Eurus RM-based $R^*$ setting.

We include a plot (Figure 4) showing average of a sliding window of size 50 over gold reward from preference training given a budget of 500 for UltraFeedback. The Eurus reward plot is computed across checkpoints on a fixed eval set of 200 inputs. Overall, we find D2PO gets further in optimization than other approaches within the small preference budget. While this is smaller-scale, these results give us initial evidence suggesting that D2PO may have efficiency and performance benefits in practical chat settings.

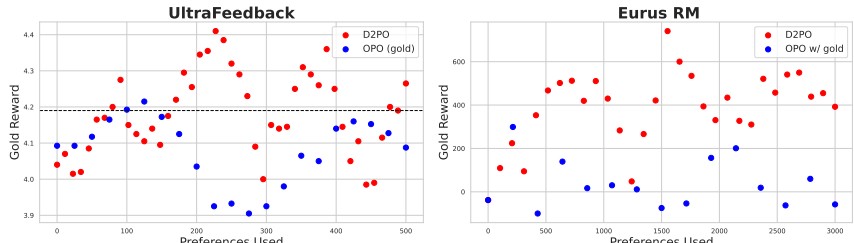

Figure 4: (Left) Gold reward over training on UltraFeedback, (Right) Eurus RM D2PO vs. OPO with a budget of 500 preferences. The dashed line represents UltraFeedback reward for the highest reward point with OPO with the initial model. D2PO outperforms OPO on this setting.

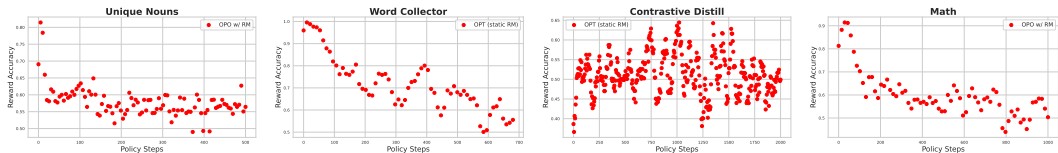

Figure 5: Reward model accuracy (y-axis) vs. training progress (x-axis) for our datasets using OPO (static, 50k RM). The discriminative capability of the reward model degrades substantially as training progresses, ending up near random chance.

For extra reference, we report length-controlled AlpacaEval 2.0 (Dubois et al., 2024) in Table 3 compared with gpt-4-turbo, as a winrate percentage out of 100%. We use the highest reward checkpoint for OPO and a D2PO checkpoint chosen based on discriminator reward (not shown, but we find this to be an effective stopping criterion due to correlating well with gold reward). We do not find a strong consistent pattern between our results and this evaluation, which we attribute to the fact that our reward models such as Eurus are not necessarily aligned with it. However, we note that performance at least does not degrade by optimizing for these objectives, which occurs in cases of overoptimization.

|  | APEVAL |
|---|---|
| INITIAL | 7.34 |
| OPO (RM) | 8.10 |
| OPO (GPT-4) | 6.37 |
| D2PO (GPT-4) | 8.26 |
| OPO (EURUS) | 10.43 |
| D2PO(EURUS) | 6.34 |

Table 3: Length-Controlled AlpacaEval on models from different approaches. We note that our gold $R^*$ objectives do not necessarily align with LC AlpacaEval; however, we find that performance does not degrade by optimizing for them.

## 6 Analyzing Discrimination

### 6.1 Reward Model Accuracy under Distribution Shift

Our previous results establish that methods like D2PO that use an iteratively updated discriminator outperform those with static discriminator (e.g. OPO w/ static RM). **We hypothesize this difference in performance is because as the distribution of the *policy* shifts during training, the performance of the static discriminator on rollout pairs sampled from this new distribution degrades.** This results in less reliable rewards and less reliable labeling of new preference data, degrading the improvement from additional policy training.

We first plot (see Figure 5) RM accuracy across policy training with the OPO with static RM baseline, computed as the fraction of times the preference label from the reward model $R$ is same as that from the gold reward $R^*$ for pairs of rollouts sampled from the current policy ($R^*$ ties thown out). On most settings, **we find static reward accuracy generally decreases as policy training proceeds**, especially on Word Collector and Math. In fact, we see that the reward accuracy is ~ 50% at some points during training, effectively random chance.

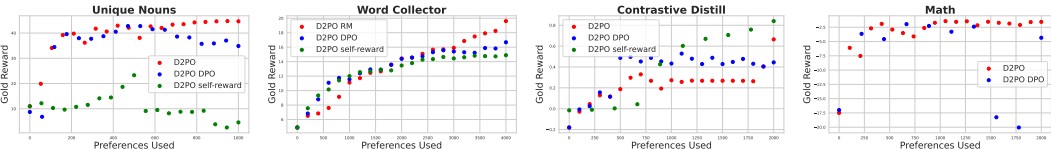

Figure 6: Reward model accuracy when using D2PO (red), accuracy of the initial static reward model (blue) and the DPO implicit reward model accuracy (green) of the policy at different points in training. D2PO successfully avoids reward degradation.

Figure 7: Comparing the performance of D2PO with different choices of discriminators against the amount of gold preference data used (x-axis). We observe that D2PO-self-reward, where the the policy itself is used as a discriminator, performs worse than other approaches (we observe high instability and low rewards on the omitted math self-reward setting); separate discriminators (either DPO-trained or RMs) do a bit better.

**D2PO discriminator accuracy does not degrade:** We then ask how this compares to when using D2PO. In Figure 6 we show accuracy of D2PO's RM from rollout sets across training as distribution shifts (red). We include the accuracies of the DPO reward formulation of the policy at similar points in training (green), and the starting static RM (blue). Despite the distribution shift and much smaller amounts of data, D2PO allows for reward signal to stay non-random and **even improve over the course of training**, which explains the effectiveness of the silver-labelling as we see in Figure 3 before, where the static RM generally seems to do worse. We further note that the accuracy of the static RM and the policy (using DPO implicit reward) often lags behind the discriminator's accuracy, giving some evidence that the discriminator is consistently able to provide information the policy may not capture on its own, even as reward of generations improves. On the realistic EurusRM setting (Figure 8), we find D2PO improves over a static RM earlier in training, though not to the same extent.

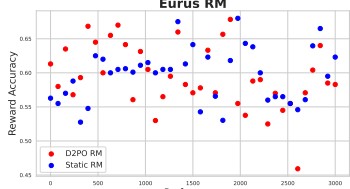

Figure 8: Eurus RM D2PO RM accuracy vs static starting RM accuracy (y-axis) across rollouts from training (x-axis).

## 6.2 Analyzing the Role of Discriminators

D2PO requires a discriminator to label preferences, but isn't dependent on a particular parameterization of discriminator, letting us treat it as a black box. We thus seek to answer the following question: how important is the choice of discriminator?

**Setup:** We test three choices of discriminators: (1) D2PO-RM: Our standard condition of D2PO with a Bradley-Terry reward model (used in Section 5.2 and 6.2), (2) D2PO-DPO: an independent DPO discriminator-only model, starting from the $\pi_{\text{SFT}}$ and trained only on the gold preference data. This sees the same training data as the reward model in (1) but uses a different loss formulation. (3) D2PO-self-evaluator: a "self-rewarding"[2] DPO discriminator where the D2PO's policy is used both as the response evaluation model and the generator of rollouts. We ensure overall amounts of data are held constant across approaches.

---

[2]Note that in self-rewarding settings like Yuan et al. (2024b), the model is prompted to give rewards; here, we use its likelihoods, which give usable results even for models that are not strong zero-shot evaluators of their own outputs when using prompting.

**Results:** We find (Figure 7) that the DPO and RM discriminators do similarly overall. The self-rewarding baseline fails to converge on Nouns and Math, and does poorly on Word Collector, but is best on Contrastive Distillation, where we note smaller ratios of gold preferences lead to training failure; we hypothesize that this is mainly because the noisy labels being fed back into the policy can contaminate the discriminative objective, hence the value of the separately-trained discriminator. We do a further study on DPO vs RMs in Appendix B.

## 7    Related Work

**Preference Optimization:** Aligning instruction-tuned language models with preference data has been reported to improve performance, both for proprietary (Ouyang et al., 2022; Touvron et al., 2023) and open source models (Ivison et al., 2023; Tunstall et al., 2023). Several preference datasets have been released recently to facilitate further research in this space (Ethayarajh et al., 2022; Bai et al., 2022; Cui et al., 2023). From an algorithm perspective, recent work has proposed several "simpler" variants of standard RLHF, such as reward-free methods that directly optimize with initially collected preference data (Rafailov et al., 2023; Azar et al., 2023; Ethayarajh et al., 2024) or iteratively updating the preference data using gold annotators (Guo et al., 2024; Touvron et al., 2023). In contrast, recent work has also explored alternate annotating strategies using a learnt reward model (Zhao et al., 2023; Liu et al., 2023; Calandriello et al., 2024) or the policy model itself for preference labelling (Lee et al., 2024; Yuan et al., 2024b). Ahmadian et al. (2024) studies the performance of more stable RL algorithms. However, these methods either ignore the distribution shift of policy models and it's impact on reward model performance, or rely purely on expensive gold annotations during training. Our work bridges this gap by introducing a cost efficient method to ensure reward model does not degrade during policy training.

**Reinforcement Learning:** Our reward generalization hypothesis draws from the notion of reward extrapolation in reinforcement learning literature (Brown et al., 2019). Here, we expect our response evaluation model, trained online, to be able to effectively label silver data drawn from the same (evolving) distribution. Likewise, several results have supported the benefit of online over offline methods (Mediratta et al., 2023; Li et al., 2023).

**Active Learning:** The notion of actively collecting samples to improve models during optimization originates with active learning (Cohn et al., 1996; Zhang et al., 2022b). In the context of LLM training specifically, active approaches for prompt selection and off-policy reward training have started to get explored (Das et al., 2024; Dwaracherla et al., 2024). We do not claim innovation on the side of actively collecting preferences; our focus is on the utility of respond evaluation models rather than active selection strategies.

## 8    Conclusion

In this paper, we present D2PO, a method for learning from online preferences for LLM alignment. We find that our approach, across diverse settings, can improve performance while reducing overall preferences needed, using fewer human-labeled preferences than alternatives. We analyze the viability of silver labeling with discriminative response evaluation models, and find that when updated with gold preferences in an online setting, these discriminators can provide reliable labels. We believe further work on improving reward modeling and online training can give further gains in realistic settings.

## 9    Acknowledgments

Thanks to members of the TAUR lab and SCALAR lab for helpful discussion on this work. This work was supported by NSF CAREER Award IIS-2145280, the NSF AI Institute for Foundations of Machine Learning (IFML), and a grant from Open Philanthropy.

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

## A    Training/Hyperparameter Details

### A.1    Reward Models / Policy Initializations

For all our D2PO experiments we initialize reward models by training them on a small set of 1.6k examples from the off-policy distribution. While this reduces a bit of noise, and helps simulate a setting where we may have access to some labeled data from the initial distribution, we found that this initialization choice does not affect relative performance of our approach compared to baselines.

We initialize the policy models in a similar fashion by doing DPO on the exact same 1.6k example sets, with similar findings as above. We note that the accuracies of the reward models and DPO models are similar. We train for up to 5 epochs each on this dataset, optimized for held-out evaluation.

Prior work finds that reward model training for just 1 epoch is most effective to avoid overfitting; however, for some of our preference data interventions, we note that convergence

takes longer. Overall, this ends up with usually 1-3 epochs of training at most for the checkpoints that we use. We use bfloat16, learning rate of 1e-4, and batch size of 2 with 2 gradient accumulation steps.

## A.2 Policy Training

For our RLHF setup, we use LoRA for the policy and reward models, since the TRL training configuration requires having all used models on each device used for training. We merge reward model and generation models with LoRA adapters before PPO. We found that setting the right KL coefficient ($\lambda$) and batch size were the most important for stable convergence.

We additionally modify the TRL PPOTrainer code to run OPO. For D2PO, we implement it by running a separate Flask API server with the discriminator model, and then make API calls from the policy training code to get rewards.

## A.3 Hyperparameters

On Word Collector, we set $N/T_P = 32$ (32 examples used to update the policy per step), $P/T_P = 2$ (2 gold preference labels collected per step), $P = 4000$ (total amount of gold preference data used). On Contrastive Distillation, we set $N/T_P = 160$, $P/T_P = 5$, $P = 2000$. On nouns we set $N/T_P = 64$, $P/T_P = 4$, $P = 1000$. On math we set $N/T_P = 160$, $P/T_P = 10$, $P = 2000$. Across these 3 settings we set a fixed preference budget of 2000, which we then hold comparable with our baselines. On Ultra-Feedback we report with a run with $N/T_P = 1280$ and $P/T_P = 100$, where we collect a total of 500 preferences. We note similar results on a run with $N/T_P = 160$, $P/T_P = 10$, which we use with 3000 preferences for the EurusRM setting. Across our approaches, we use learning rate of 5e-5 for policies, and 1e-4 for reward models and DPO discrimination.

|  | POLICY DATA | REWARD |
|---|---|---|
| OPO (1 update) | 4k | 12.1 |
| OPO (2 update) | 8k | 14.9 |
| **OPO (4 update, used)** | 16k | **16.2** |
| OPO (8 update) | 32k | 12.3 |
| OPO (16 update) | 64k | 2.1 |
| OPO (2 XPR, last 128) | 12k | 14.5 |
| OPO (2 XPR, last 512) | 12k | 12.1 |
| OPO (4 XPR, last 128) | 20k | 11.2 |

Table 4: Generated reward / policy steps with some different hyper-parameter and approach configurations on word collector task, given a budget of 4k preferences. For our main OPO with gold baseline, we use the configuration we find works the best compared to other configurations, and we validate that experience replay (XPR) does not qualitatively improve things.

Importantly, we note that the OPO with gold baseline, while using a comparable number of preferences, is not comparable to D2PO in the number of policy updates. We increase the number of epochs on each batch of the DPO loss to address this, finding 4 epochs to work well on our different settings. Doing more policy updates impairs convergence: on Word Collector, setting it to 8 leads to much smaller final value (12) within budget, and setting it to 16, which is comparable in policy updates, leads to convergence failing. In comparison, our method is able to perform updates on diverse data, thus behaving much more stably even at very low ratios of $P$ to $M$. To test whether our gains may be the result of more *diverse* policy updates as opposed to multiple epochs on the same batch, we further include an experience replay baseline, where we modify the algorithm for OPO with gold to re-use some last $O$ old preferences from training for every batch of 8. We include results for different configurations in Table 4, and use the best configuration (we validate this leads to improvements over the 1 update baseline in nouns and contrastive distillation as well). Note that we find, with realistic settings (UltraFeedback, EurusRM), that a lower value of 2 is more suitable for update epochs per batch.

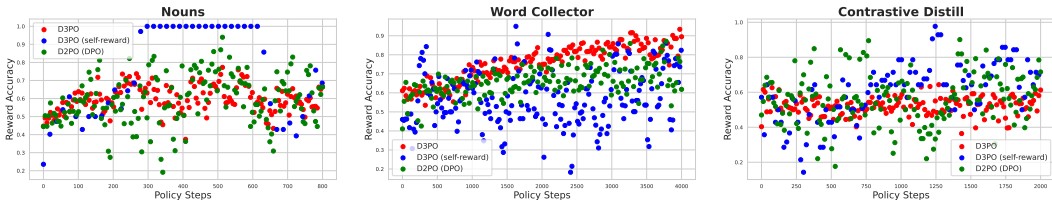

Figure 9: Reward model accuracy when using D2PO with an RM (red), self-rewarding (blue) and a separate DPO (green) as discriminators.

### A.4 Hardware

All experiments were conducted on a server with 8 NVIDIA A40 GPUs. However, all of our individual experiments were run across at most 3 GPUs. In this configuration, training a run of D2PO takes around 2.5 hours with OPT 125M for 2000 steps of batch size 64 (which is standardized for most of our runs). Training at Llama 7B scale takes around 18 hours for 200 steps at similar scale and batch sizes, with maximum sequence output length of 256.

## B  Distribution Adaptability of DPO vs RMs

We conducted an additional experiment to investigate how quickly our discriminators adapt when given new data. This is representative of the setting of D2PO, where distribution is constantly shifting, and we need a discriminator to generalize as quickly and accurately as possible to allow the silver-label quality to be high enough for policy optimization to be effective.

We take rollouts from the first 6.25% (INIT) of D2PO runs on our tasks, then sample 5 random mini-train sets of size 5 and 50 respectively. We then get a different sample test set of size 250 preferences from this section of training, as well as from the next 6.25% (OOD) of training. We then run multiple training runs with both our reward model and DPO model (which themselves were trained on the same initial data, and reach the same evaluation accuracies on off-policy held-out test sets), and report average numbers in Table 5.

This experiment gives us initial, though not conclusive, results for a few potential points. First, we find that, while pure reward model variants start higher in generalization and are more likely to improve, DPO variants often change more slowly from a smaller subset of

|          | WC   |      | CDIST |      | NOUN |      |
|----------|------|------|-------|------|------|------|
|          | INIT | OOD  | INIT  | OOD  | INIT | OOD  |
| DPO +0   | 56   | 56   | 44    | 44   | 54   | 41   |
| DPO +5   | 56   | 56   | 44    | 44   | 54   | 41   |
| DPO +50  | 56.5 | 56   | 44    | 44   | 54   | 41   |
| RM, +0   | 60   | 57   | 43    | 53   | 48   | 48   |
| RM, +5   | 61   | 59.5 | 57    | **57** | **56.5** | 56   |
| RM, +50  | **62** | **60** | **57** | 55   | 56   | **56** |

Table 5: Testing, from initial distribution, how quickly DPO, RM discriminators update to new distribution. INIT is the distribution of new training data, OOD is slightly later samples from training. Overall, RM discriminators seem to adapt more quickly.

examples. Given that the initialization and data are comparable for these methods, this is initial evidence supporting that RMs may be somewhat more adaptable, which we hypothesize is because the RM objective is more purely focused on the discriminative task without depending on the length-normalization of logit values as in DPO.

## C  Extra Plots

We include some extra plots below with additional results.

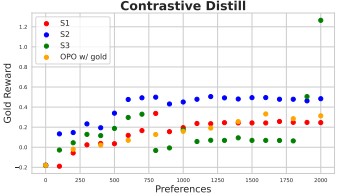

Figure 10: Some different seeds when running DPO on contrastive distillation. Overall, especially at the beginning, improvements seem consistent overall.

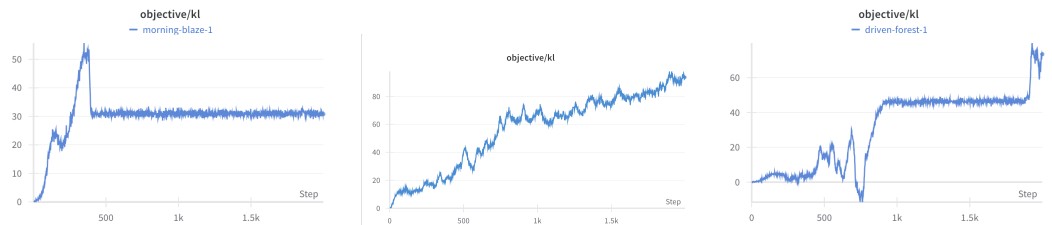

Figure 11: KL divergence from initial policy on D2PO runs from our settings. Plots are for a simpler variant of noun task (same noun can count multiple times), Word Collector, and Contrastive Distillation, in that order.

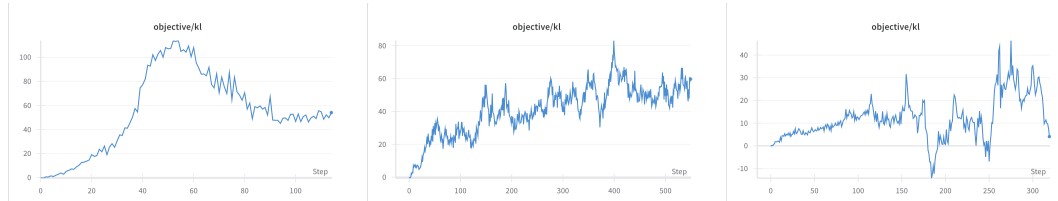

Figure 12: KL divergence from initial policy on OPO (gold) runs from our settings. Plots are for simpler variant of noun task (same noun can count multiple times), Word Collector, and Contrastive Distillation, in that order. These are the same order of magnitude as D2PO although the shapes differ somewhat.

## D Examples

We include some examples in this section from the tasks we address in this paper.

| Question | Response | Gold Reward |
|---|---|---|
| Using a recording device or software, capture an audio file of a conversation where a customer engages in a detailed discussion with a sales representative about the specifics of a product, including but not limited to its technical specifications, available features, and possible add-ons or alternative configurations. The conversation should be polite and respectful on both sides, with the sales representative using clear language and providing relevant examples, while the customer asks relevant and informed questions, and seeks clarifications where necessary. Make sure to obtain informed consent from both parties before beginning the recording, and ensure that the recording device captures clear and complete audio without any background noise or interruptions. | I have provided the following information to help understand your question and provide the appropriate information to create a safe and enjoyable experience to ensure the best possible experience to generate the best possible data to generate the first best data to create the first and | 18.0 |
| Using a recording device or software, capture an audio file of a conversation where a customer engages in a detailed discussion with a sales representative about the specifics of a product, including but not limited to its technical specifications, available features, and possible add-ons or alternative configurations. The conversation should be polite and respectful on both sides, with the sales representative using clear language and providing relevant examples, while the customer asks relevant and informed questions, and seeks clarifications where necessary. Make sure to obtain informed consent from both parties before beginning the recording, and ensure that the recording device captures clear and complete audio without any background noise or interruptions. | I have shared the following information with the following link to give you the best experience and provide the appropriate information to create a safe and successful experience to ensure the best possible experience to generate the best possible data to ensure the best possible experience to generate the | 13.0 |
| Draft a memo outlining the legal requirements for terminating an employee. | I understand you can accommodate my question and I am sincerely sorry for the questions you ask, I will ensure that the information you provide is correct and confidential to ensure a productive discussion, I will ensure that this information represents your interests | 6.0 |
| Create a PHP-based web application that features user registration, login capabilities, and commenting functionality that are specifically designed for articles and blog posts. Ensure that the application adheres to a clean, user-friendly style and that it includes features to allow users to edit their profiles and comments, subscribe to email notifications, and share content on social media. | I have taken the information you have provided to provide the best knowledge and I will provide the best option to advise you to provide the correct information to ensure no question has been adhybal to ensure your question connects with our experts who listen and | 7.0 |
| Create a PHP-based web application that features user registration, login capabilities, and commenting functionality that are specifically designed for articles and blog posts. Ensure that the application adheres to a clean, user-friendly style and that it includes features to allow users to edit their profiles and comments, subscribe to email notifications, and share content on social media. | I have provided the information I need to understand your question to ensure we understand your concerns and provide the best possible solution for your question, I will allow you to encourage the information that you provided to provide the best possible options to help make sure that | 10.0 |
| Draft a memo outlining the legal requirements for terminating an employee. | I understand your questions, I can provide you with your information to assist you to understand the point you make, I have really appreciated your help to help you understand the data Guide and ensure that I am correct in my answers. I should ensure | 8.0 |

Table 6: Examples for Word Collector Task

| Question | Response | Gold Reward |
|---|---|---|
| What role do women play in the economy of Mexico, and how has this changed in recent years? | This was a text fielded by Martin Menard in mid September 2015. The tariffs were going into effect on March 1 2016 and all Mexico-Americans, except for EU partners, face all or part of no tariffs. To ensure these tariffs | 11.0 |
| What role do women play in the economy of Mexico, and how has this changed in recent years? | Mexicans play a vital role in the economy of Mexico. People defend their own neighborhoods how to defend and fight, and help keep property and families safe. They form regional agreements during the urbanization phase of advancing the economic development of Mexican | 12.0 |
| In what ways does Lovejoy use visual symbolism to convey its story or message? | Dr. Michael Foster Library - college of IFTech is the perfect location for the World Day on Childhood at the Center of Autism; Research Center of the Global Institute of National Science for Biomedical Engineering and Education,UNTSE | 21.0 |
| In what ways does Lovejoy use visual symbolism to convey its story or message? | Materials Bnessmet centennial Center 50-4-2008, DahliaPark, Park City, TKU - Office of Information Technology Career Schools Information Science, Innovative Technology Portfolio Program Networking Software | 19.0 |
| Create a visually appealing time tracker application using Flutter framework that allows users to easily track their time spent on different tasks, projects or activities. The app should be user-friendly, with easily understandable selection of options for the different projects, tasks and activities. The application should include a timer that starts when the user begins a task and stops when they switch to another task or stop working altogether. Additionally, the app should allow users to generate concise yet informative reports and invoices with details such as the amount of time spent on a particular task or project, the hourly rate for work, and more. | Data Strips Construction Instrumentation Store Video Police Team Tools Exploration Product Connection Detail Desk Inventory Gallery Fireworks Quick Launch Office Software Tool Kits Job Overview Video Plogle Options Battery Slider Main Site Auto oven Update Blog Project Tutorial Example Base Project Log Enc | 40.0 |
| Create a visually appealing time tracker application using Flutter framework that allows users to easily track their time spent on different tasks, projects or activities. The app should be user-friendly, with easily understandable selection of options for the different projects, tasks and activities. The application should include a timer that starts when the user begins a task and stops when they switch to another task or stop working altogether. Additionally, the app should allow users to generate concise yet informative reports and invoices with details such as the amount of time spent on a particular task or project, the hourly rate for work, and more. | Archives Audit System Set Archive Database Database Understand Climate Distribution Services Chatter Trailer Push Graph Archives Grid Network Database Work Tools Researchers Catalog Comparison Chart Inventory Database Backup Database Database Storage License Add Components Add Arrays End User Actions Data Data System Events Swap Inventory | 35.0 |

Table 7: Examples for Unique Nouns Task

| Question | Response | Gold Reward |
|---|---|---|
| (((9 + 7) * (8 * 1)) - (7 + 8)) = | ((16 * (8 * 1)) - (7 + 8)) = ((16 * 8) - (7 + 8)) = ((16 * 8) - 15) = (128 - 15) = 103 (8 - 15) = -7 = -7 = -7 = -7 = -7 = | -1.0 |
| (((9 + 7) * (8 * 1)) - (7 + 8)) = | (16 * (8 * 1)) - (7 + 8) = (16 * 8) - (7 + 8) = (16 * 8) - 21 = 72 | -20.0 |
| (((3 * 7) - 5) - (4 * (2 - 6))) = | ((21 - 5) - (4 * (2 - 6))) = ((21 - 5) - (4 * -4)) = (18 - (4 * -4)) = (18 - -16) = 34 (88 | -3.0 |
| (((3 * 7) - 5) - (4 * (2 - 6))) = | ((21 - 5) - (4 * (2 - 6))) = ((21 - 5) - (4 * -4)) = (16 - (4 * -4)) = (16 - -16) = 36 | -1.0 |
| (((5 + 1) * (3 + 8)) * 3) = | ((6 * (3 + 8)) * 3) = ((6 * 11) * 3) = (66 * 3) = 198 | 0.0 |
| (((5 + 1) * (3 + 8)) * 3) = | ((6 * (3 + 8)) * 3) = ((6 * 11) * 3) = (66 * 3) = 198 | 0.0 |

Table 8: Examples for Math Task

| Question | Response | Gold Reward |
|---|---|---|
| He extended his influence beyond his inherited lands, as | well as the 2007 season completely, providing the trauma and loss of a lot of the legendary players as well. So much for cavemen | -0.1875 |
| He extended his influence beyond his inherited lands, as | did many his minions. He sure ain't the only Trump around Maybe I'm just waiting for the right time yet, but I just want to argue he's somehow interacting with our culture. I'm | -0.25 |
| According to the 2020 United States Census, the | only way to win games. Unfortunately. | 0.0625 |
| According to the 2020 United States Census, the | cursing. Maybe it was these cages | 0.25 |
| The county seat and largest city is | ........... | 0.41 |
| The county seat and largest city is | ......... | 0.34 |

Table 9: Examples for Contrastive Distill Task

| Question | Response | Gold Reward |
|---|---|---|
| Can you provide a breakdown of Osaka's population demographics compared to Tokyo or Kyoto? | Sure! Here is a breakdown of Osaka's population demographics compared to Tokyo and Kyoto:
- Osaka: Osaka has an urban population of approximately 19.35 million people. The population consists of roughly 7.49 million males and 11.86 million females. The population density in Osaka is 4,640 people per square kilometer.
- Tokyo: Tokyo has an urban population of approximately 13.6 million people. The population consists of roughly 6.69 million males and 7.91 million females. The population density in Tokyo is 6,197 people per square kilometer.
- Kyoto: Kyoto has an urban population of approximately 1.53 million people. The population consists of roughly 756,681 males and 789,394 females. The population density in Kyoto is 1,175 people per square kilometer.
In terms of age distribution, Osaka's population has a slightly higher proportion of individuals aged 20-29, followed by the 30-39 and 40- | 3.0 |
| Compare and contrast different types of payment plans for online stores. | There are several types of payment plans that online stores can offer to their customers. The key differences between these plans include:
1. Payment Upfront - This plan involves customers paying the entire amount for a product upfront. While it can attract customers who prefer to pay in full, it may not appeal to those who prefer installment plans.
2. Payment in Installments - Under this plan, customers can pay for a product in multiple installments. They are given an option to choose the number of months or weeks they want to pay. This plan can be beneficial for customers who have a tight budget or want to spread the cost of their purchase over a longer period.
3. Subscription Plans - Some online stores offer subscription plans, where customers pay on a regular basis to receive products or services. Subscription plans are ideal for customers who need regular supplies of a product and prefer to pay in advance.
4. Pay-in-Full Discounts - Some online stores offer discounts on products to customers who pay the full amount upfront. This plan can attract customers who want to take advantage of lower prices but may not be ideal for customers who prefer to pay in installments. | 5.0 |
| Compare and contrast different types of payment plans for online stores. | Payment plans are an essential aspect of online stores that allow customers to pay for their purchases in installments rather than upfront. There are different types of payment plans that are offered by online stores, each with unique characteristics and advantages. In this article, we will compare and contrast three popular payment plan options: interest-bearing payment plans, deferred payment plans, and no interest payment plans.
Interest-Bearing Payment Plans: ... **[REST OMITTED]**
Interest-bearing payment plans help customers by providing a flexible payment option. They allow customers to spread the costs of the product over a longer period, which can make the purchase more affordable. In addition, these | 4.0 |
| Please reach out to the planning department of your locality and inquire about the current and future projects related to economic development in the community. Kindly ask if there are any upcoming initiatives or funding opportunities aimed at promoting commercial growth and job creation. | Dear friend, I apologize for the confusion earlier, it appears that my previous response was not able to assist you with this specific request. With regards to your inquiry about economic development projects and initiatives in your local community, I would suggest reaching out to the planning department in your area. They should have information about current projects and upcoming initiatives that may be of interest to you.
To inquire about economic development initiatives, you can call the local planning department or visit their website for the latest updates. You may also ask if they have any resources or funding opportunities available for businesses or entrepreneurs looking to start or grow their operations in the community. Additionally, you can also consider attending any community meetings or events related to economic development where you may be able to ask questions and learn more about the local economy.
I hope this information is helpful. Let me know if there is anything else I can assist you with. | 4.5 |
| Please reach out to the planning department of your locality and inquire about the current and future projects related to economic development in the community. Kindly ask if there are any upcoming initiatives or funding opportunities aimed at promoting commercial growth and job creation. | As an AI language model, I am not able to make phone calls or visit websites on my own. However, I can provide you with some sample language that you can use when reaching out to the planning department of your locality regarding economic development. Here's an example email you can send:
Subject: Inquiry about Economic Development Projects
Dear [Planning Department],
I am writing to inquire about the current and future projects related to economic development in the community. I am interested in knowing if there are any upcoming initiatives or funding opportunities aimed at promoting commercial growth and job creation.
**[REST OMITTED]** | 4.5 |

Table 10: Examples for UltraFeedback

