# OpenReview forum: "D2PO: Discriminator-Guided DPO with Response Evaluation Models"
_colmweb.org/COLM/2024/Conference — COLM_

### Official Review · Reviewer_v8LM · 2024-05-11

**Rating:** 6
**Confidence:** 4
**Ethics Flag:** 1

**Summary:**

This paper propose D2PO, discriminator-guided DPO,
an approach for the online setting where preferences are being collected throughout learning. By collecting golden preference annotations during online sampling, D2PO simutenously trains the policy and the reward models.

**Reasons To Accept:**

- The idea is quite potential, and the designed synthetic experiments are good.
- The comparison of different discriminators is worth presenting.

**Reasons To Reject:**

- What is OPO and how is it implemented? OPO is not a well-established term in RLHF. I went through the two cited papers. Liu et al. use rejection sampling to collect preference pairs with the policy model, then apply DPO or SLiC loss for preference optimization. Lee et al. proposes a self-judgment approach where the reward model and policy model are initialized together. So it is not clear what OPO is referring to in this paper. From Figure 1, I guess OPO first trains an RM with offline preference data, then labels on-policy data with the static RM. However, it is necessary to clarify the loss function and specific implementation details.
- The efficiency issue. For real tasks, the cost of acquiring on-policy gold labels would be high (e.g. human annotation). So the feasibility of this method is questionable.
- The online RL methods such as PPO did not appear as a baseline, which also updated value models during training, but did not require gold labels.
- There are three synthetic experiments with only one real-world experiment. It would be better to test the method on more real benchmarks. For example, you can try math or coding problems where the gold labels are easy to get.

Overall, I think the main assumption of this work, better utilization of on-policy data, would make contribution to the field. I will be happy to increase my score once I see more convincing real-world experiments.

---

> ### Author Rebuttal · Authors · 2024-05-28
>
> Thanks for the thoughtful suggestions/comments!
>
> > Online methods such as PPO not included
>
> We do include a PPO baseline in Table 2, where we generally find PPO to be less stable and performant than OPO with the same data conditions. As mentioned in the paper, we also try PPO for D2PO, which doesn’t work at all. PPO optimizes a scalar, which is tricky to optimize when the reward is constantly updated.
>
> > more real benchmarks, e.g. math/coding
>
> Thanks for suggesting! We ran 3 new settings to address this:
>
> **Realistic**: We use EurusRM, the top ranked on reward bench, as gold reward (R*) instead of GPT-4. We find that online optimization and D2PO both lead to strong improvements. We include a comparison plot for D2PO and OPO w/ gold here: https://ibb.co/fHL2qvK , which we will include in any future version of the work.
>
> **Math / reasoning**: Given a math expression, generate a chain of steps to solve the output (eg. “((1+2)+4) = (3+4) = 7”). R* is based on the correctness of each step. D2PO performs similarly to the OPO baseline, but online preferences largely improve over offline (DPO fails, similar to nouns). D2PO RM accuracy plots have more noisy accuracy compared to other settings, which may explain this.
>
> **Improvements to the Noun task**: To avoid trivial repetition, we test a version where only the unique # of nouns count. The results show the same trends as the original noun setting, with D2PO optimizing faster, but methods max out at 43 instead of 50.
>
> We’ll include updated results for these settings in any future version.
>
> > What's OPO
>
> OPO (Section 3/3.1) involves iteratively (1) sampling a small set of paired rollouts from policy; (2) labeling pairs using some discriminator; (3) doing some updates (DPO loss function; epochs per batch tuned) on this batch of preferences. OPO w/ static RM uses a frozen RM for labels in (2), where OPO w/ gold uses the gold reward R*. Similar prior algorithms exist (e.g. Lee et al), but add extra complexity. OPO is a unified template that captures the core design choices of this prior work more comparably to D2PO. We will clarify this in any future version.
>
> > Efficiency
>
> This is actually the main motivation of this paper, which seeks to find ways to improve the efficiency of preference data collection. Online preference data collection is standard in modern LLM training (e.g., Llama 2). We argue that these schemes can be improved by working in conjunction with silver-labeling such as with our proposed algorithm.

---

> > ### Comment · Reviewer_v8LM · 2024-06-05
> > **Reply**
> >
> > Thanks for your reply. I think the additional results and explanation are beneficial. I will maintain my score and lean positive to the paper.

---

### Official Review · Reviewer_kQQH · 2024-05-12

**Rating:** 4
**Confidence:** 4
**Ethics Flag:** 1

**Summary:**

While direct preference optimization (DPO) gains popularity, it is often performed on static preference data collected ahead of the training, limiting its theoretical capability. This paper proposes D2PO, an online version of DPO that annotates preference data on-the-fly during training through an iteratively updated reward model. To achieve this, D2PO requires periodically collecting gold preference labels and retraining the reward model. Under a fixed budget, the authors show that D2PO achieves the best performance over several baselines on several diverse tasks.

**Questions To Authors:**

* The “discriminator” in D2PO is actually the reward or preference model, but this term is very confusing in the context of LLM alignment.
* In Section 6.1, the authors argue that the updated reward model improves the reward accuracy. However,  the accuracy in Contrastive Distill doesn’t change that much comparing Figure 5 and 6. Could you please explain why D2PO still outperforms DPO w/ RM?
* In Figure 3 and 4, why does OPO w/ gold reward model underperforms D2PO? If D2PO benefits from improved reward accuracy, gold rewards always give perfect reward for the training, but OPO w/ gold reward performs even worse.
* With a fixed budget, how does the iteration T_p affect the performance? Should we perform more iterations, each annotating a smaller amount of gold preference data, or fewer iterations?
* It’s not very clear how gold preference and silver preference data are used during the training, and why they have different amounts as shown in Table 1.

**Reasons To Accept:**

The proposal of D2PO that achieves promising alignment performance on several tasks, demonstrates the significance of online and on-policy learning for LLM alignment

**Reasons To Reject:**

- Clarity should be improved.
- Claims are not always supported by the experimental results.
- One advantage of DPO is to drop the separate reward model which greatly simplifies the training pipeline. D2PO, however, brings it back and relies on iteratively updated reward modeling, making the whole pipeline more complicated.

---

> ### Author Rebuttal · Authors · 2024-05-28
>
> Thanks for the thoughtful comments and feedback!
>
> > “discriminator”
>
> The reason we used a more general term here was because we wanted to note that there are different types of models (self-reward, reward model, likelihood under generative model) which can be used interchangeably to get silver preference labels. We will clarify the usage of this term in a future version.
>
> > Section 6.1
>
> Thanks for pointing this out! Note that Figures 5 and 6 aren’t comparable, since one uses outputs from OPO w/ static RM (much lower gold reward), and the other uses D2PO training outputs, where the RM is updated during training. When testing static RM accuracy on the Figure 6 set, we found that it is lower overall than the D2PO RM, and more noisy, often dropping substantially at certain training steps in training. While online updates help D2PO’s RM handle these unfamiliar steps’ distributions, OPO w/ RM gets stuck or even degrades further at these points, hence doing worse. We will include this static RM comparison in future versions of Figure 6.
>
> > OPO w/ gold reward underperforms D2PO in Figure 3 and 4
>
> This actually follows our main hypothesis. Note OPO w/ gold has access to the same number of gold preferences, but doesn’t use silver preferences (see Table 1). When OPO w/ gold is beaten by D2PO, this thus indicates that the additional silver preferences, which we can only obtain with a separate discriminator, are actually helpful and lead to more efficient optimization.
>
> > how does the iteration T_p affect the performance?
>
> We generally found that, given the same amount of preferences, collecting less data more frequently generally improves performance, since the data stays more on-distribution with policy.
>
> > Gold preference vs silver data during training
>
> Gold / silver preferences are treated the same and both are used to update the policy model, whereas the discriminator model is updated with only gold preferences as they’re collected. Silver data is much cheaper since we can get this for “free” if we have some sort of discriminator, which is why we usually have much more silver data than gold data. This is what gives D2PO an advantage in preference data efficiency.
>
> > complexity of D2PO
>
> We agree that there are more moving parts in this approach, but our argument is that there are benefits to bringing back the reward model. Note that because we use LoRA adapters for training, the additional memory cost and complexity of juggling additional models is minimal.

---

> > ### Comment · Reviewer_kQQH · 2024-06-05
> >
> > Thanks for the response!
> >
> > Re "discriminator"
> >
> > In my opinion, the term "reward or preference" model explains things well. I didn't see the necessity of using "discriminator" here.
> >
> > Re "Section 6.1"
> >
> > Without comparing to Figure 5, Figure 6, the Contrastive Distill particularly, doesn't show clear benefit from online updates. It would be great to explain why the policy model still improves in this condition.
> >
> > Re "T_p"
> >
> > Please include the corresponding experimental results.
> >
> > Re "OPO w/ gold"
> >
> > In the paper, the authors explain that OPO (gold): only online gold preference labels are used to update the policy. Doesn't OPO (gold) adopts perfect reward function during training since it's online? I may misunderstand gold/silver preference data, but as I mentioned before, clarity needs improved.
> >
> > I prefer to keep my original judgement.

---

> > > ### Author Response · Authors · 2024-06-06
> > > **Continued Response**
> > >
> > > Thanks for following up! We definitely agree that clarity of the original version needs improvement, and have been working on a much clearer version (especially on explanations of the algorithm and data condition) based on your feedback.
> > >
> > > > Section 6.1
> > >
> > > We also found it interesting that contrastive distillation seems to improve despite having lower overall reward accuracy than other tasks. It’s worth noting that in standard RLHF, even on large datasets, RMs are known to have low (~60%) accuracy.  Our hypothesis is that the noisy features reward models learn do, in aggregate, correlate enough with the true objective for optimization to succeed, and the online setting helps this feature learning in aggregate. Referring to Figure 3 (comparable with Fig 6), note that the beginning and the end of policy training (where Fig 6 shows highest stable accuracy) is where policy improvement occurs the most, which makes sense.
> > >
> > > > T_p
> > >
> > > We can include more complete results in a future version. Some reference results (high T_p means more online):
> > >
> > > Word collector:
> > > - T_p=2000 ->  max R ~=20
> > > - T_p=500 ->  max R ~=17.9
> > > - T_p=8 ->  max R ~=15.3
> > >
> > > Contrastive Distill:
> > > - T_p=400 ->  max R ~=0.7
> > > - T_p=20 ->  max R ~=0.2
> > >
> > > Nouns:
> > > - T_p=200 -> max R ~=49 (with 200 gold prefs)
> > > - T_P=20 -> max R ~=49 (with 600 gold prefs)
> > >
> > >  > OPO w/ gold
> > >
> > > This is an important thing we’ve tried to improve clarity on in our new version. We can explain further: for example, in OPO w/ gold, we may sample 32 preferences in a step, and label these with the gold reward function. With D2PO, we also get 32 gold labels, and can update the policy on these, but since we have an RM, we can *also* generate 128 (or more) new outputs on fresh prompts, and get more preference labels using the RM, without needing any new gold labels. This thus gives DPO its advantage in gold preference data efficiency.

---

### Official Review · Reviewer_Gtpm · 2024-05-19

**Rating:** 7
**Confidence:** 4
**Ethics Flag:** 1

**Summary:**

Current reinforcement learning approaches in alignment learning of LLMs are fundamentally constrained by the inability of reward models, which are generally trained with an initial set of human-labeled gold preferences, to generalize to the shifting distributions of policy models, often leading to susceptibility to reward hacking and bias towards superfluous features such as the response length. The paper offers a thorough investigation of the discriminator settings when the budget for online preference labeling is limited, and thus contributing to the field of online RL. Specifically, the paper proposes a training framework that alternates between training the discriminator/reward model using fresh data sampled from the latest policy model annotated with gold preferences and training the policy model with the dynamically updated reward model. The paper also proposes training the policy model with DPO in each alternating training step. The experiment results on synthetic problems such as writing-with-keywords and noun maximization and on real datasets (UltraChat) support the claim that the proposed method D2PO (essentially an online variant of DPO with a dynamically adjusted reward model trained with a fixed total budget of gold preference labeling) enables the policy model to reach the target reward more efficient and effectively.

**Questions To Authors:**

What seed prompts did you use for generating responses to label with the reward model in each self-training step?

**Reasons To Accept:**

- The paper is well-motivated and provides sufficient background to justify its research statement and implications.
- The experimental settings, including the synthetic and real-world datasets, are well-designed to support their claim. The results are also promising, providing more weight on online RL.
- The method's key differentiations with the closely related work, "self-rewarding language models" (Yuan et al., 2024), include the incorporation of gold preference labeling throughout the online training and the discriminator setting (whether to be unified with the policy model, etc.). They provide a unique perspective on online RLHF, and the findings will be very useful for AL practitioners.

**Reasons To Reject:**

- The paper fails to explore when the budget on gold preference labeling is larger -- e.g., investigating whether having more initial gold preferences diminishes the benefit of subsequent iterative online training with silver preferences.

---

> ### Author Rebuttal · Authors · 2024-05-28
>
> Thanks for the thoughtful comments and feedback!
>
> > e.g., investigating whether having more initial gold preferences diminishes the benefit of subsequent iterative online training with silver preferences.
>
> This is a good question. With respect to offline gold preference budget for the initial policy model, as mentioned in the paper we note we don’t find major differences in downstream performance for the full D2PO pipeline with different preference budgets. We explore variants with up to 50k initial offline preferences (Table 2, stars in Figure 3) and don’t find a big difference from the 1k preference setting. In general, the value of initial preferences vs. the value of online preferences varies by setting, but online preferences do always give benefit over offline preferences (compare DPO vs online approaches in Figure 3).
>
> > seed prompts
>
> Thanks for clarifying this! We briefly mention this in Section 4.1: for Word Collector, Nouns, and UltraFeedback we use a subsampled set of prompts from the UltraFeedback dataset. For the contrastive distillation we use truncated sentences (5-15 tokens) from the wikidata dataset. We will clarify this in the paper.

---

### Decision · Program_Chairs · 2024-07-10

**Decision:**

Accept

**Comment:**

## Summary
A limitation of most LLM alignment is that it is done as offline RL on fixed preference datasets. This means the reward models do not necessarily reflect the shifting distribution of the trained policy model. This paper presents an online version of DPO with a dynamically adjusted silver-standard reward, D2PO, which might be simpler and more efficient than, but as effective as, PPO. The clarity of the paper could be better. The idea seems right in many circumstances if perhaps an obvious direction and implementation. I didn't really understand kQQH's low rating. This seemed to me a pretty good if not revolutionary paper, which it would be useful to see published.
## Strengths
* Well-motivated
* Good experiments
* Additional experiments added during rebuttal.
## Weaknesses
* Only explores the case of limited, fixed budget training.
* Negates the simplicity of DPO by returning to more complex reward+policy.
* The paper should have more comparison with PPO. (But difficulties getting PPO to work are discussed.)